# OUT-OF-SAMPLE EXTRAPOLATION WITH NEURON EDITING

## ABSTRACT

While neural networks can be trained to map from one specific dataset to another, they usually do not learn a generalized transformation that can extrapolate accurately outside the space of training. For instance, a generative adversarial network (GAN) exclusively trained to transform images of black-haired men to blond-haired men might not have the same effect on images of black-haired women. This is because neural networks are good at generation within the manifold of the data that they are trained on. However, generating new samples outside of the manifold or extrapolating "out-of-sample" is a much harder problem that has been less well studied. To address this, we introduce a technique called *neuron editing* that learns how neurons encode an edit for a particular transformation in a latent space. We use an autoencoder to decompose the variation within the dataset into activations of different neurons and generate transformed data by defining an editing transformation on those neurons. By performing the transformation in a latent trained space, we encode fairly complex and non-linear transformations to the data with much simpler distribution shifts to the neuron's activations. We motivate our technique on an image domain and then move to our two main biological applications: removal of batch artifacts representing unwanted noise and modeling the effect of drug treatments to predict synergy between drugs.

## 1 INTRODUCTION

Many experiments in biology are conducted to study the effect of a treatment or a condition on a set of samples. For example, the samples can be groups of cells and the treatment can be the administration of a drug. However, experiments and clinical trials are often performed on only a small subset of samples from the entire population. Usually, it is assumed that the effects generalize in a context-independent manner without mathematically attempting to model the effect and potential interactions with the context. However, mathematically modeling the effect and potential interactions with background information would give us a powerful tool that would allow us to assess how the treatment would generalize beyond the samples measured.

We propose a neural network-based method for learning a general edit function corresponding to treatment in the biological setting. While neural networks offer the power and flexibility to learn complicated ways of transforming data from one distribution to another, they are often overfit to the training dataset in the sense that they only learn how to map one specific data manifold to another, and not a general edit function. Indeed, popular neural network architectures like GANs pose the problem as one of learning to *generate* the post-treatment data distributions from pre-treatment data distributions. Instead, we reframe the problem as that of learning an *edit* function between the pre- and post- treatment versions of the data, that could be applied to other datasets.

We propose to learn such an edit, which we term *neuron editing*, in the latent space of an autoencoder neural network with non-linear activations. First we train an autoencoder on the entire population of data which we are interested in transforming. This includes all of the pre-treatment samples and the post-treatment samples from the subset of the data on which we have post-treatment measurements.

The internal layers of this autoencoder represent the data with all existing variation decomposed into abstract features (neurons) that allow the network to reconstruct the data accurately (Vincent et al., 2008; Baldi, 2012; Le, 2013; Shin et al., 2012). Neuron editing involves extracting differences between the observed pre-and post-treatment activation distributions for neurons in this layer and

then applying them to pre-treatment data from the rest of the population to synthetically generate post-treatment data. Thus performing the edit node-by-node in this space actually encodes complex multivariate edits in the ambient space, performed on denoised and meaningful features, owing to the fact that these features themselves are complex non-linear combinations of the input features.

While neuron editing is a general technique that could be applied to the latent space of any neural network, even GANs themselves, we instead focus exclusively on the autoencoder in this work to leverage three of its key advantages. First, we seek to model complex distribution-to-distribution transformations between large samples in high-dimensional space. While this can be generally intractable due to difficulty in estimating joint probability distributions, research has provided evidence that working in a lower-dimensional manifold facilitates learning transformations that would otherwise be infeasible in the original ambient space (Zhu et al., 2016; Patel et al., 2013; Vincent et al., 2010). The non-linear dimensionality reduction performed by autoencoders finds intrinsic data dimensions that essentially *straighten* the curvature of data in the ambient space. Thus complex effects can become simpler shifts in distribution that can be computationally efficient to apply.

Second, by performing the edit to the neural network internal layer, we allow for the modeling of some context dependence. Some neurons of the internal layer have a drastic change between pre- and post-treatment versions of the experimental subpopulation, while other neurons such as those that encode background context information not directly associated with treatment have less change in the embedding layer. The latter neurons are less heavily edited but still influence the output jointly with edited neurons due to their integration in the decoding layers. These edited neurons interact with the data-context-encoding neurons in complex ways that may be more predictive of treatment than the experimental norm of simply assuming widespread generalization of results context-free.

Third, editing in a low-dimensional internal layer allows us to edit on a denoised version of the data. Because of the reconstruction penalty, more significant dimensions are retained through the bottleneck dimensions of an autoencoder while noise dimensions are discarded. Thus, by editing in the hidden layer, we avoid editing noise and instead edit significant dimensions of the data.

We note that neuron editing makes the assumption that the internal neurons have semantic consistency across the data, i.e., the same neurons encode the same types of features for every data manifold. We demonstrate that this holds in our setting because the autoencoder learns a joint manifold of all of the given data including pre- and post-treatment samples of the experimental subpopulation and pre-treatment samples from the rest of the population. Recent results show that neural networks prefer to learn patterns over memorizing inputs even when they have the capacity to do so (Zhang et al., 2016).

We demonstrate that neuron editing extrapolates better than generative models on two important criteria. First, as to the original goal, the predicted change on extrapolated data more closely resembles the predicted change on interpolated data. Second, the editing process produces more complex variation, since it simply preserves the existing variation in the data rather than needing a generator to learn to create it. We compare the predictions from neuron editing to those of several generation-based approaches: a traditional GAN, a GAN implemented with residual blocks (ResnetGAN) to show generating residuals is not the same as editing (Szegedy et al., 2017), and a CycleGAN (Zhu et al., 2017). While in other applications, like natural images, GANs have shown an impressive ability to generate plausible individual points, we illustrate that they struggle with these two criteria. We also motivate why neuron editing is performed on inference by comparing against a regularized autoencoder that performs the internal layer transformations during training, but the decoder learns to undo the transformation and reconstruct the input unchanged (Amodio et al., 2018).

In the following section, we detail the neuron editing method. Then, we motivate the extrapolation problem by trying to perform natural image domain transfer on the canonical CelebA dataset (Liu et al., 2018). We then move to two biological applications where extrapolation is essential: correcting the artificial variability introduced by measuring instruments (batch effects), and predicting the combined effects of multiple drug treatments (combinatorial drug effects) (Anchang et al., 2018).

## 2 MODEL

Let $S, T, X \subseteq \mathbb{R}^d$ be sampled sets representing $d$-dimensional source, target, and extrapolation distributions, respectively. We seek a transformation that has two properties: when applied to $S$

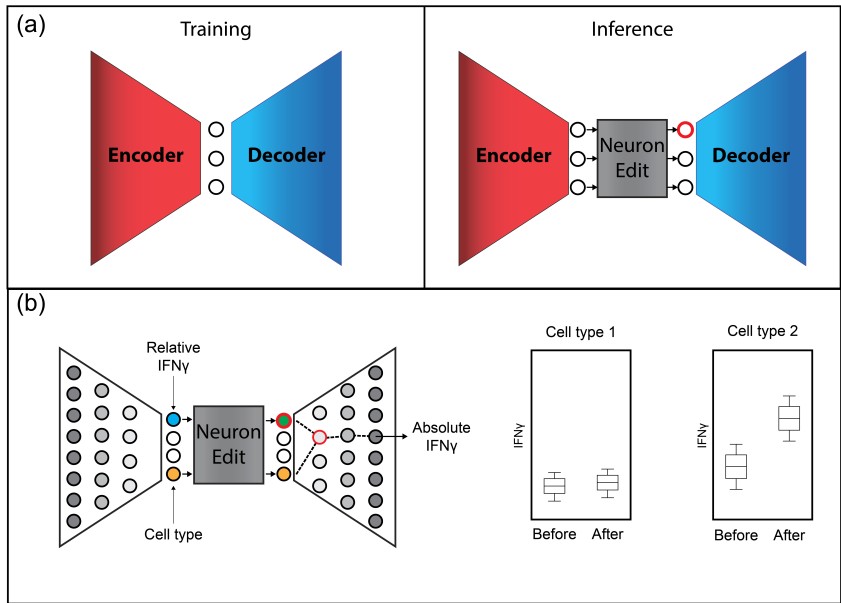

Figure 1: (a) Neuron editing interrupts the standard feedforward process, editing the neurons of a trained encoder/decoder to include the source-to-target variation (Equation 1), and letting the trained decoder cascade the resulting transformation back into the original data space. (b) The network can incorporate context information, like cell type, that is encoded in other neurons as changes induced by the editing transformation cascade through the decoder.

it produces a distribution equivalent to the one represented by $T$, and when applied to $T$ it is the identity function. GANs learn a transformation with these properties, and when parameterized with ReLU or leaky ReLU activations, as they often are, this transformation also has the property that it is piecewise linear (Goodfellow et al., 2014; Kadurin et al., 2017). However, the GAN optimization paradigm produces transformations that do not behave comparably on both $S$ and $X$. Therefore, instead of *learning* such a transformation, we *define* a transformation with these properties on a *learned* space (summarized in Figure 1).

We first train an encoder/decoder pair $E/D$ to map the data into an abstract neuron space decomposed into high-level features such that it can also decode from that space, i.e., an objective $L$:

$$L(S, T, X) = \text{MSE}\left[(S, T, X), D(E(S, T, X))\right]$$

where MSE is the mean-squared error. Then, without further training, we separately extract the activations of an $n$-dimensional internal layer of the network for inputs from $S$ and from $T$, denoted by $a_S : S \to \mathbb{R}^n, a_T : T \to \mathbb{R}^n$. We define a piecewise linear transformation, called $NeuronEdit$, which we apply to these distributions of activations:

$$NeuronEdit(a) = \begin{cases} \left(\frac{a - p_0^S}{p_1^S - p_0^S} \cdot (p_1^T - p_0^T)\right) + p_0^T & a < p_1^S \\ \left(\frac{a - p_j^S}{p_{j+1}^S - p_j^S} \cdot (p_{j+1}^T - p_j^T)\right) + p_j^T & a \in [p_j^S, p_{j+1}^S) \\ \left(\frac{a - p_{99}^S}{p_{100}^S - p_{99}^S} \cdot (p_{100}^T - p_{99}^T)\right) + p_{99}^T & a \geq p_{99}^S \end{cases} \tag{1}$$

where $a \in \mathbb{R}^n$ consists of $n$ activations for a single network input, $p_j^S, p_j^T \in \mathbb{R}^n$ consist of the $j^{th}$ percentiles of activations (i.e., for each of the $n$ neurons) over the distributions of $a_S, a_T$ correspondingly, and all operations are taken pointwise, i.e., independently on each of the $n$ neurons in the layer. Then, we define $NeuronEdit(a_S) : S \to \mathbb{R}^n$ given by $x \mapsto NeuronEdit(a_S(x))$, and equivalently for $a_T$ and any other distribution (or collection) of activations over a set of network inputs. Therefore, the $NeuronEdit$ function operates on distributions, represented via activations over network input samples, and transforms the input activation distribution based on the difference between the source and target distributions (considered via their percentile disctretization).

We note that the $NeuronEdit$ function has the three properties of a GAN generator: 1. $NeuronEdit(a_S) \approx a_T$ (in terms of the represented $n$-dimensional distributions) 2. $NeuronEdit(a_T) = a_T$ 3. piecewise linearity. However, we are also able to guarantee that the neuron editing performed on the source distribution $S$ will be the same as that performed on the extrapolation distribution $X$, which would not be the case with the generator in a GAN.

To apply the learned transformation to $X$, we first extract the activations of the internal layer computed by the encoder, $a_X$. Then, we cascade the transformations applied to the neuron activations through the decoder without any further training. Thus, the transformed output $\hat{X}$ is obtained by:

$$\hat{X} = D(NeuronEdit(E(X)))$$

Crucially, the nomenclature of an *autoencoder* no longer strictly applies. If we allowed the encoder or decoder to train with the transformed neuron activations, the network could learn to undo these transformations and still produce the identity function. However, since we freeze training and apply these transformations exclusively on inference, we turn an autoencoder into a generative model that need not be close to the identity.

Training a GAN in this setting could exclusively utilize the data in $S$ and $T$, since we have no real examples of the output for $X$ to feed to the discriminator. Neuron editing, on the other hand, is able to model the variation intrinsic to $X$ in an unsupervised manner despite not having real post-transformation data for $X$. Since we know *a priori* that $X$ will differ substantially from $S$, this provides significantly more information.

Furthermore, GANs are notoriously tricky to train (Salimans et al., 2016; Gulrajani et al., 2017; Wei et al., 2018). Adversarial discriminators suffer from oscillating optimization dynamics (Li et al., 2017), uninterpretable losses (Barratt & Sharma, 2018; Arjovsky et al., 2017), and most debilitatingly, mode collapse (Srivastava et al., 2017; Kim et al., 2017; Nagarajan & Kolter, 2017). Mode collapse refers to the discriminator being unable to detect differences in variabillity between real and fake examples. In other words, the generator learns to generate a point that is very realistic, but produces that same point for most (or even all) input, no matter how different the input is. In practice, we see the discriminator struggles to detect differences in real and fake distributions even without mode collapse, as evidenced by the generator favoring ellipsoid output instead of the more complex and natural variability of the real data. Since our goal is not to generate convincing *individual examples* of the post-transformation output, but the more difficult task of generating convincing *entire distributions* of the post-transformation output, this is a worrisome defect of GANs.

Neuron editing avoids all of these traps by learning an unsupervised model of the data space with the easier-to-train autoencoder. The essential step that facilitates generation is the isolation of the variation in the neuron activations that characterizes the difference between source and target distributions.

There is a relationship between neuron editing and the well-known word2vec embeddings in natural language processing (Goldberg & Levy, 2014). There, words are embedded in a latent space where a meaningful transformation such as changing the gender of a word is a constant vector in this space. This vector can be learned on one example, like transforming *man* to *woman*, and then extrapolated to another example, like *king*, to predict the location in the space of *queen*. Neuron editing is an extension in complexity of word2vec's vector arithmetic, because instead of transforming a single point into another single point, it transforms an entire distribution into another distribution.

## 3 EXPERIMENTS

In this section we compare neuron interference as an editing method to various generating methods: a regularized autoencoder, a standard GAN, a ResnetGAN, and a CycleGAN. For the regularized autoencoder, the regularization penalized differences in the distributions of the source and target in a latent layer using maximal mean discrepancy (Amodio et al., 2018; Dziugaite et al., 2015). The image experiment used convolutional layers with stride-two filters of size four, with 64-128-256-128-64 filters in the layers. All other models used fully connected layers of size 500-250-50-250-500. In all cases, leaky ReLU activation was used with $0.2$ leak. Training was done with minibatches of size 100, with the adam optimizer (Kingma & Ba, 2014), and a learning rate of $0.001$.

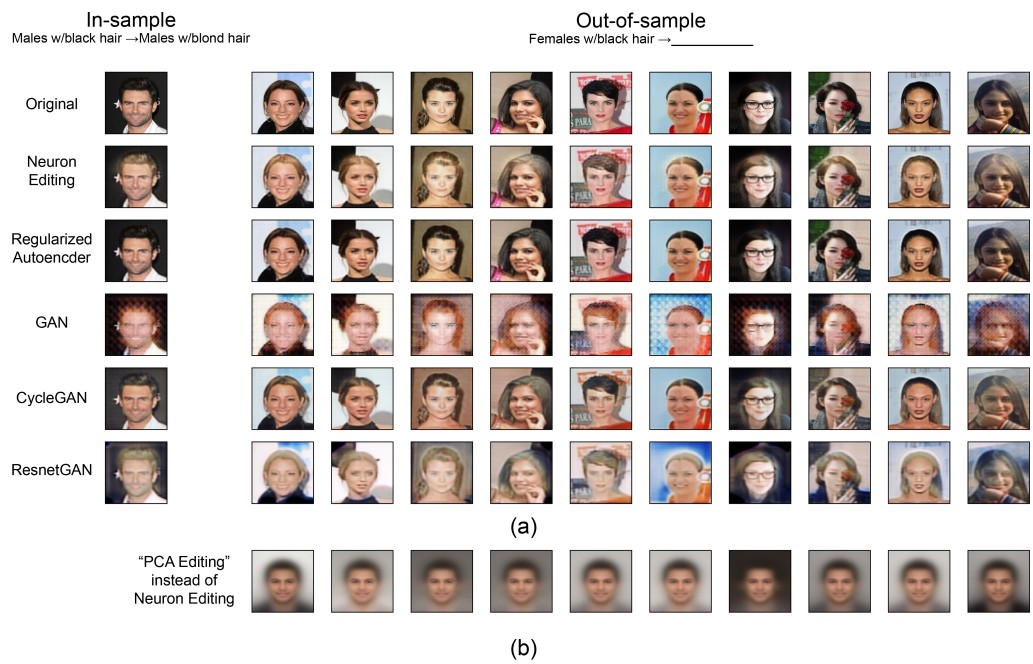

Figure 2: Data from CelebA where the source data consists of males with black hair and the target data consists of males with blond hair. The extrapolation is then applied to females with black hair. (a) A comparison of neuron editing against other models. Only neuron editing successfully applies the blond hair transformation. (b) An illustration that neuron editing must be applied to the neurons of a deep network, as opposed to principle components.

### 3.1 CelebA Hair Color Transformation

We first consider a motivational experiment on the canonical image dataset of CelebA (Liu et al., 2018). If we want to learn a transformation that turns a given image of a person with black hair to that same person except with blond hair, a natural approach would be to collect two sets of images, one with all black haired people and another with all blond haired people, and teach a generative model to map between them. The problem with this approach is that the learned model will only be able to apply the hair color change to images similar to those in the collection, unable to extrapolate.

This is illustrated in Figure 2a, where we collect images that have the attribute male and the attribute black hair and try to map to the set of images with the attribute male and the attribute blond hair. Then, after training on this data, we extrapolate and apply the transformation to females with black hair, which had not been seen during training. The GAN models are unable to successfully model this transformation on out-of-sample data. In the parts of the image that should stay the same (everything but the hair color), they do not always generate a recreation of the input. In the hair color, only sometimes is the color changed. The regular GAN model especially has copious artifacts that are a result of the difficulty in training these models. This provides further evidence of the benefits of avoiding these complications when possible, for example by using the stable training of an autoencoder and editing it as we do in neuron editing.

In Figure 2b, we motivate why we need to perform the $NeuronEdit$ transformation on the internal layer of a neural network, as opposed to applying it on some other latent space like PCA. Only in the neuron space has this complex and abstract transformation of changing the hair color (and only the hair color) been decomposed into a relatively simple and piecewise linear shift.

### 3.2 Batch correction by out-of-sample extension from Spike-in Samples

We next demonstrate another application of neuron editing's ability to learn to transform a distribution based on a separate source/target pair: batch correction. Batch effects are differences in the

observed data that are caused by technical artifacts of the measurement process. In other words, we can measure the same sample twice and get two rather different datasets back. When we measure different samples, batch effects get confounded with the true difference between the samples. Batch effects are a ubiquitous problem in biological experimental data that (in the best case) prevent combining measurements or (in the worst case) lead to wrong conclusions. Addressing batch effects is a goal of many new models (Finck et al., 2013; Tung et al., 2017; Butler & Satija, 2017; Haghverdi et al., 2018), including some deep learning methods (Shaham et al., 2017; Amodio et al., 2018).

One method for grappling with this issue is to repeatedly measure an identical control (spike-in) set of cells with each sample, and correct that sample based on the variation in its version of the control (Bacher & Kendziorski, 2016). In our terminology of generation, we choose our source/target pair to be Control1/Control2, and then extrapolate to Sample1. Our transformed Sample1 can then be compared to raw Sample2 cells, rid of any variation induced by the measurement process. We would expect this to be a natural application of neuron editing as the data distributions are complex and the control population is unlikely to be similar to any (much less all) of the samples.

The dataset we investigate in this section comes from a mass cytometry (Bandura et al., 2009) experiment which measures the amount of particular proteins in each cell in two different individuals infected with dengue virus (Amodio et al., 2018). The data consists of 35 dimensions, where Control1, Control2, Sample1, and Sample2 have 18919, 22802, 94556, and 55594 observations, respectively. The two samples were measured in separate runs, so in addition to the true difference in biology creating variation between them, there are also technical artifacts creating variation between them. From the controls, we can see one such batch effect characterized by artificially low readings in the amount of the protein InfG in Control1 (the x-axis in Figure 3a).

We would like our model to identify this source of variation and compensate for the lower values of InfG without losing other true biological variation in Sample1. For example, Sample1 also has higher values of the protein CCR6, and as the controls show, this is a true biological difference, not a batch effect (the y-axis in Figure 3a). Because the GANs never trained on cells with high CCR6 and were never trained to generate cells with high CCR6, it is unsurprising that all of them remove that variation. Worryingly, the GAN maps almost all cells to the same values of CCR6 and InfG (Figure 3d), and the CycleGAN maps them to CCR6 values near zero (Figure 3f). This means later comparisons would not only lose the information that these cells were exceptionally high in CCR6, but now they would even look exceptionally low in it. The ResnetGAN does not fix this problem, as it is intrinsic to the specification of the generation objective, which only encourages the production of output like the target distribution, and in this case we want output different from the target distribution. As in the previous examples, the ResnetGAN also learns residuals that produces more ellipsoid data, rather than preserving the variation in the original source distribution or even matching the variation in the target distribution. The regularized autoencoder has learned to undo the transformations to its latent space and produced unchanged data.

Neuron editing, on the other hand, decomposes the variability into just the separation between controls (increases in InfG), and edits the sample to include this variation. This removes the batch effect that caused higher readings of InfG, while preserving all other real variation, including both the intra-sample variation and the variation separating the populations in CCR6.

Unlike the other generative models, neuron editing produced the intended transformations for the proteins InfG and CCR6, and here we go on to confirm that its results are accurate globally across all dimensions. In Figure 4a, a PCA embedding of the whole data space is visualized for Control1 (light blue), Control2 (light red), Sample1 (dark blue), and post-transformation Sample1 (dark red). The transformation from Control1 to Control2 mirrors the transformation applied to Sample1. Notably, the other variation (intra-sample variation) is preserved. In Figure 4b, we see that for every dimension, the variation between the controls corresponds accurately to the variation introduced by neuron editing into the sample. These global assessments across the full data space offer additional corroboration that the transformations produced by neuron editing reasonably reflect the transformation as evidenced by the controls.

### 3.3 COMBINATORIAL DRUG TREATMENT PREDICTION ON SINGLE-CELL DATA

Finally, we consider biological data from a combinatorial drug experiment on cells from patients with acute lymphoblastic leukemia (Anchang et al., 2018). The dataset we analyze consists of cells

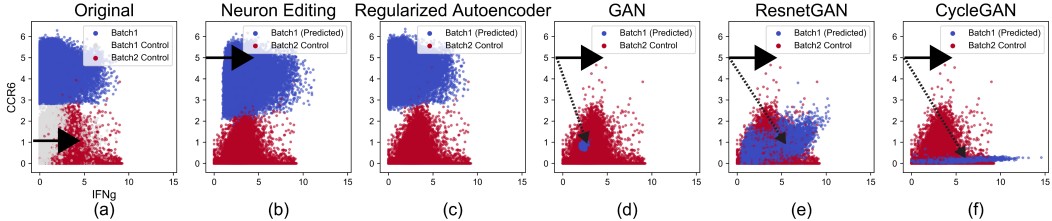

Figure 3: The results of learning to batch correct a sample based on the difference in a repeatedly-measured control population measured. The true transformation is depicted with a solid arrow and the learned transformation with a dashed arrow. There is a batch effect that should be corrected in IFNg (horizontal) with true variation that should be preserved in CCR6 (vertical). All of the GANs attempt to get rid of all sources of variation (but do so only partially because the input is out-of-sample). The autoencoder does not move the data at all. Neuron editing corrects the batch effect in IFNg while preserving the true biological variation in CCR6.

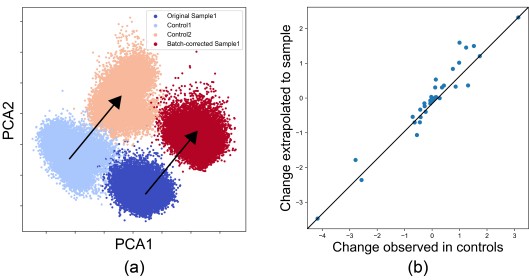

Figure 4: (a) The global shift in the two controls (light blue to red) is isolated and this variation is edited into the sample (dark blue to red), with all other variation preserved. (b) The median change in the sample in each dimension corresponds accurately with the evidence in each dimension in the controls.

under four treatments: no treatment (basal), BEZ-235 (Bez), Dasatinib (Das), and both Bez and Das (Bez+Das). These measurements also come from mass cytometry, this time on 41 dimensions, with the four datasets consisting of $19925$, $20078$, $19843$, and $19764$ observations, respectively. In this setting, we define the source to be the basal cells, the target to be the Das cells, and then extrapolate to the Bez cells. We hold out the true Bez+Das data and attempt to predict the effects of applying Das to cells that have already been treated with Bez.

A characteristic effect of applying Das is a decrease in p4EBP1 (seen on the x-axis of Figure 5). No change in another dimension, pSTATS, is associated with the treatment (the y-axis of Figure 5). Neuron editing accurately models this horizontal change, without introducing any vertical change or losing variation within the extrapolation dataset (Figure 5a). The regularized autoencoder, as before, does not change the output at all, despite the manipulations within its internal layer (Figure 5b). None of the three GAN models accurately predict the real combination: the characteristic horizontal shift is identified, but additional vertical shifts are introduced and much of the original within-Bez variability is lost (Figure 5c-e).

We note that since much of the variation in the target distribution already exists in the source distribution and the shift is a relatively small one, we might expect the ResnetGAN to be able to easily mimic the target. However, despite the residual connections, it still suffers from the same problems as the other model using the generating approach: namely, the GAN objective encourages all output to be like the target it trained on.

That the GANs are not able to replicate even two-dimensional slices of the target data proves that they have not learned the appropriate transformation. But to further evaluate that neuron editing produces a meaningful transformation globally, we here make a comparison of every dimension. Figure 6 compares the real and predicted means (in 6a) and variances (in 6b) of each dimension. Neuron editing more accurately predicts the principle direction and magnitude of transformation across all dimensions. Furthermore, neuron editing better preserves the variation in the real data. In almost all dimensions, the GAN generates data with less variance than really exists.

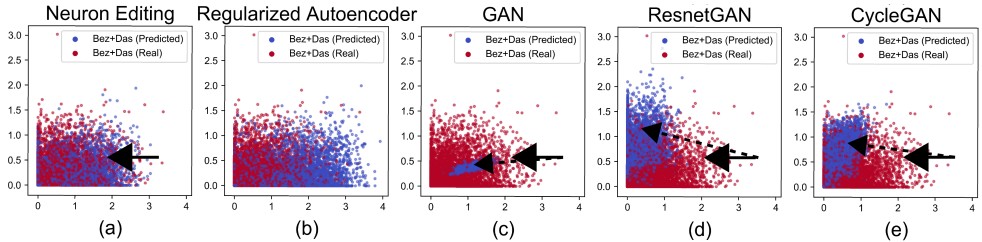

Figure 5: Combinatorial drug data showing predicted (blue) and real (red) under both Bez and Das drug treatments. The true transformation is depicted with a solid arrow and the predicted transformation with a dashed arrow. All of the GANs have trouble learning to generate the full variability of the target distribution, especially from out-of-sample data. The autoencoder does not even make the slight transformation necessary, and instead leaves the data unchanged. Neuron editing results in meaningful edits that best models the true combination.

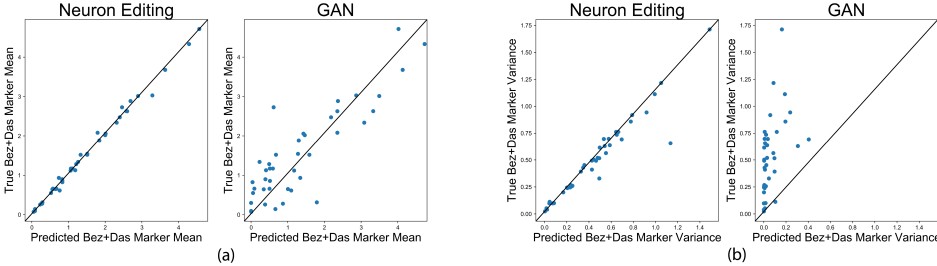

Figure 6: A comparison of real and predicted means and variances per dimension. Neuron editing more accurately predicts the change in mean than the GAN and better preserves the true variance than the GAN. The GAN almost uniformly generates lower variance data than really exist.

## 4 DISCUSSION

In this paper, we tackled a data-transformation problem inspired by biological experimental settings: that of generating transformed versions of data based on observed pre- and post-transformation versions of a small subset of the available data. This problem arises during clinical trials or in settings where effects of drug treatment (or other experimental conditions) are only measured in a subset of the population, but expected to generalize beyond that subset. Here we introduce a novel approach that we call *neuron editing*, for applying the treatment effect to the remainder of the dataset. Neuron editing makes use of the encoding learned by the latent layers of an autoencoder and extracts the changes in activation distribution between the observed pre-and post treatment measurements. Then, it applies these same edits to the internal layer encodings of other data to mimic the transformation. We show that performing the edit on neurons of an internal layer results in more realistic transformations of image data, and successfully predicts synergistic effects of drug treatments in biological data. Moreover, we note that it is feasible to learn complex data transformations in the non-linear dimensionality reduced space of a hidden layer rather than in ambient space where joint probability distributions are difficult to extract. Finally, learning edits in a hidden layer allows for interactions between the edit and other context information from the dataset during decoding. Future work along these lines could include training parallel encoders with the same decoder, or training to generate conditionally.

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
