# OpenReview forum: "Out-of-Sample Extrapolation with Neuron Editing"
_ICLR.cc/2019/Conference_

### Official Review · AnonReviewer3 · 2018-11-02
**Interesting idea in improving data transformation generalization across input data distributions in an unsupervised way**

**Rating:** 6
**Confidence:** 4

**Review:**

The authors proposed a novel method of making data transformation that is much easier to extend to the cases where the input distribution is different from the one that is used to the train the model (in-sample vs out-sample). This has a lot of application in removing experimental noise in biological data (also known as batch effects).
The idea is to learn a representation that separates background (dimensions that do NOT vary across data points, but may be subject to change in a data transformation) and foreground (dimensions that vary between data points under the same background) and then apply a *fixed* linear transformation in the learned representation space. This is different from other approaches, such as GAN, where the transformation is learned entirely based on the data. In addition, it mitigates some known problems, such as the "mode collapse" in GAN, by just learning a good representation. This is proposed to be done by an autoencoder trained on both in-samples and out-samples (the transformation is however adjusted based on the in-samples only). Experimental results are appealing in different applications compared to GAN, ResnetGAN, and CycleGAN.
Here are my major concerns:
- The idea seems to be very general and indeed is applicable to any latent representation learning method, and not just autoencoders. Is there any reason that other more complicated unsupervised representation learning methods were not used for benchmarking in the paper?

- The method heavily relies on the quality of the unsupervised learned representation. How one is guaranteed that the transformation in the learned space be simple and piecewise linear? Shouldn't we consider a regularization method to guide the unsupervised learning more appropriately?

- The method also implicitly assumes that the same neurons model background and foreground in the in-sample and out-sample data points. How is that guaranteed in practice?

---

> ### Author Response · Authors · 2018-11-26
> **We thank the reviewer for the recommendation of acceptance, and we appreciate the reviewer’s strong grasp of the major ideas we introduce in the paper**
>
> We thank the reviewer for the recommendation of acceptance, and we appreciate the reviewer’s strong grasp of the major ideas we introduce in the paper. However, we emphasize that the goal of the paper is not to specifically learn an image-domain transformation (like background to foreground), but to learn a generic transformation that could apply to background/foreground, or drug perturbations, or any other application, without needing to be re-defined with the particulars of each case.
>
> We would like to address the feedback provided in-line:
>
> ---------------------------------
> Here are my major concerns:
> - The idea seems to be very general and indeed is applicable to any latent representation learning method, and not just autoencoders. Is there any reason that other more complicated unsupervised representation learning methods were not used for benchmarking in the paper?
> ---------------------------------
> This is a good point, and as we mention in our paper, neuron editing can indeed be used on the internal representations of any type of network. We focused on autoencoders because of their proven ability to learn rich representations of the data and place them on a low-dimensional manifold, while still being very stable to train. We were able to achieve the desired transformations with autoencoders, so we felt no need to bring in the downsides and complications that more complicated networks bring along with their benefits (our paper shows many examples of GAN training difficulties like mode collapse and oscillating around the minimax equilibrium). We believe this can be a viable area of future research for interested readers.
>
>
> ---------------------------------
> - The method heavily relies on the quality of the unsupervised learned representation. How one is guaranteed that the transformation in the learned space be simple and piecewise linear? Shouldn't we consider a regularization method to guide the unsupervised learning more appropriately?
> ---------------------------------
> As we argue in the paper, the limitation that we learn a non-linear function that is still piecewise linear is not very restricting at all. Since GANs also do this, and we know they can transform one high-resolution image into another completely different high-resolution image, this class of functions is still extremely expressive. The idea of making editing easier with regularizations is an interesting one, beyond the scope of what we consider in this manuscript, but one could easily imagine one such regularization, for example, reducing the redundancy between features with a mutual information metric. We contend that neuron editing’s success without any additional restrictions on the learning is a compelling advantage.
>
>
> --------------------------------
> - The method also implicitly assumes that the same neurons model background and foreground in the in-sample and out-sample data points. How is that guaranteed in practice?
> ---------------------------------
> Indeed, the reviewer brings up another astute point. Unfortunately, as in many cases in deep learning, we cannot provide any theoretical guarantees that a neuron models the same thing in both in-sample and out-of-sample points. However, in practice, this motivates our decision to use a single unsupervised autoencoder on the entire data (both in- and out-of-sample). Since the autoencoder doesn’t have any way of distinguishing between in-sample and out-of-sample points, it is not able to systematically operate on a pseudo-label or anything differentiating them. In fact, previous work has shown that gradient descent prefers to pick up on smooth structure rather than narrow memorization [1,2].
>
> [1] Nguyen, Q., & Hein, M. (2018, July). Optimization landscape and expressivity of deep cnns. In International Conference on Machine Learning (pp. 3727-3736).
> [2] Zhou, P., & Feng, J. (2018). Understanding Generalization and Optimization Performance of Deep CNNs. arXiv preprint arXiv:1805.10767.

---

### Official Review · AnonReviewer1 · 2018-11-04
**border-line paper**

**Rating:** 5
**Confidence:** 3

**Review:**

The paper demonstrates that we can harness (semantically meaningful) features learned by a pre-trained autoencoder AE to define a determinisc transformation (e.g. math operations on latent space) to transform one distribution A into another distribution B.
The original AE was pre-trained on a larger distribution that includes both A and B.

A key contribution of this paper is the interesting demonstration that this method (called Neuron Editing) allows us to perform a transformation T that transforms  pre-treatment observations into post-treatment observations, which is useful in the medical or biological setting.

+ Novelty
Neuron editing is essentially a common technique of performing arithmetics in the latent space e.g. King - Man + Woman = Queen (in NLP) or Man wearing sunglass - Man + Woman = Woman wearing sunglasses (e.g. in image domain e.g. in Alec Radford et al. 2015).
Therefore, the novelty is limited.

+ Significance
The main contribution of this paper is the empirical demonstration that such transformation T is better defined, rather than learned directly from data (e.g. via GANs).

I should note that I'm not too familiar with the biology datasets in Sec. 3.2 and Sec 3.3 in order to fully appreciate the practical impact of Neuron Editing.

+ Clarity

I think some key reasons behind why Neuron Editing works could be more clearly presented.
That is, the key here is we use pre-trained AEs to perform a pre-defined transformation.
I think the key might not be whether we use GANs or not, it is how we use them.
I guess if we use ALI (i.e. training a GAN concurrently with an AE) to perform Neuron Editing, the result should work as well.

---

> ### Author Response · Authors · 2018-11-26
> **We thank the reviewer for the recommendation of acceptance, and appreciate the reviewer’s recognition of the contributions of our paper.**
>
> We thank the reviewer for the recommendation of acceptance, and appreciate the reviewer’s recognition of the contributions of our paper.
>
> We would like to address the feedback provided in-line:
>
> ---------------------------------
> “+ Novelty
> Neuron editing is essentially a common technique of performing arithmetics in the latent space e.g. King - Man + Woman = Queen (in NLP) or Man wearing sunglass - Man + Woman = Woman wearing sunglasses (e.g. in image domain e.g. in Alec Radford et al. 2015).
> Therefore, the novelty is limited.”
> ---------------------------------
> We disagree that the novelty is limited, as our work is distinct from all of the examples stated above in two main ways: they assume a single vector being applied to all different points, and they only look at interpolation not extrapolation.
>
> In previous works, they assume the transformation is a single vector that is global across the whole space. They create a single average estimate of a “sunglasses transformation vector” across several points and then apply this single vector globally. Neuron editing is a complex non-linear transformation over an entire distribution, rather than applying a single vector to many different points.
>
> Also, in those works, all points are interpolations from the training data, while our goal is to generate extrapolated data. To demonstrate the difference between interpolating and extrapolating, consider that in the NLP example, the words king, man, queen, and woman are all in-sample and trained on. Or, in the image example, their face dataset contained pictures of men, pictures of women, pictures of men with sunglasses, and women with sunglasses. The previous work had no obvious ability to learn the sunglass transformation on just pictures of men (with and without glasses), and then apply out-of-sample to never-before-seen pictures of women and create pictures of women with sunglasses. In our biological applications, extrapolation was essential.
>
> ---------------------------------
> "+ Significance
> The main contribution of this paper is the empirical demonstration that such transformation T is better defined, rather than learned directly from data (e.g. via GANs)."
> ---------------------------------
> We emphasize that while we define the family of functions we learn (T), the specific transformation is still learned from the data.
>
>
> ---------------------------------
> "I should note that I'm not too familiar with the biology datasets in Sec. 3.2 and Sec 3.3 in order to fully appreciate the practical impact of Neuron Editing."
> ---------------------------------
> While we understand the reviewer’s lack of familiarity, we would like to point out that nothing of this type has been done in biology before. The levels of noise, the complexity of the distributions, and the need to extrapolate from limited data have made existing generative models that were designed mostly for other uses particularly poorly suited to single-cell data. We hope the reviewer can still consider this in the final decision, as two of our three applications (especially exciting ones, no less) focus on biological applications (from the papers of Anchang et al. and Amodio et al.). If it would enhance readability, we would be happy to include a supplemental section explaining the biological concepts in greater detail.
>
>
> ---------------------------------
> "+ Clarity
> I think some key reasons behind why Neuron Editing works could be more clearly presented.
> That is, the key here is we use pre-trained AEs to perform a pre-defined transformation.
> I think the key might not be whether we use GANs or not, it is how we use them.
> I guess if we use ALI (i.e. training a GAN concurrently with an AE) to perform Neuron Editing, the result should work as well."
> ---------------------------------
> We agree, as we state in the paper, that neuron editing is a general transformation that can be performed on any neural network, even though we focused exclusively on autoencoders in our particular work. The motivation for this decision is that autoencoders have been shown to learn disentangled representation of the data in its latent space [1]. Additionally, by using autoencoders when we can, we avoid the well-known training difficulties of GANs (mode collapse, oscillating around equilibrium, unbalanced generator and discriminator, etc.). However, in principle, there would be no problem with using neuron editing with an adversarial autoencoder as the reviewer mentions, and we have incorporated this specifically to the manuscript.
>
> [1] Bengio, Y., Courville, A., & Vincent, P. (2013). Representation learning: A review and new perspectives. IEEE transactions on pattern analysis and machine intelligence, 35(8), 1798-1828.

---

> > ### Comment · AnonReviewer1 · 2018-12-02
> > **agree with most, but unclear about extrapolation ability**
> >
> > Thanks, the authors for their clarification! I agree with most comments from the author.
> >
> > However, I still am not convinced that this work advances the extrapolation ability over the previous (which is a much harder question).
> > Some of the other methods (e.g. GANs) that the authors compare with were not trained to perform feature extraction. Instead, they were trained with different objectives, and therefore it is not surprising they underperformed Neuron Editing.
> >
> > The key contribution of this work (as I see) is the demonstration of the use of a pre-trained AE, which is not too surprising to me. Although, they may have benefits in biological applications.

---

### Official Review · AnonReviewer2 · 2018-11-09
**Novel but Limited on Image Data**

**Rating:** 5
**Confidence:** 3

**Review:**

The authors present a way to transform data from a source distribution to have characteristics of a target distribution.
This is accomplished by applying a "NeuronEdit" function to the encoding of the input; this edited input is then decoded.
The NeuronEdit function is parametrized by the target distribution's statistics. The edit function does a sort of simple histogram matching, so that the ith percentile values of the source distribution's bottleneck representations instead become the ith percentile values of the target distribution's bottleneck representations.
Experiments are on CIFAR-10 and biology datasets (the latter of which are not my strong suit).

This paper is well-written and original. It is original because there are only a few works which directly manipulate the latent space (one example is latent space interpolation used to visualize GANs), and this is distinct from those.
The problem they aim to solve also has not received much attention, which enhances the novelty of this paper.
The presented method is simple and easy to implement, since the editing function is not learned but is instead deterministic. It is encapsulated in Equation 1.

The fact that the editing function is fixed may greatly hinder its flexibility and applicability.
In Section 3.1 and Figure 1, we are shown that NeuronEditting can turn images of horses with white backgrounds into images of horses with dark backgrounds (horses are an unseen class).
NeuronEditting turns the horse darker as well. It seems that one could change the brightness and contrast of the image to obtain a similar effect, or one could or take the geometric mean of the image in [0,1] with the average target image and obtain a similar effect. Such traditional methods are also robust to unseen classes. Moreover, NeuronEditting's ability to change the brightness of the image is not that surprising given that brightness is some of the most basic image information. (In point of fact it is captured by the DC Coefficient, the very first coefficient from the discrete cosine transform which is used in JPEG.) What else can NeuronEditting do in the image domain? Can this be used to rotate or reflect MNIST digits? The biological experiments also appear to involve simple input transformations.

Fine points:
- "an edit function between the the"
- I am not sure the speculation about this method's loose relation to word2vec belongs in a scientific work. Both involve modifications to a neural representations, but no further relation is justified in the paper.
- Was the dataset partitioning for the CIFAR-10 experiment done manually? If not, what process partitioned the dataset?

Edit: Some of my suggestions were incorporated in the rebuttal, but my sentiment is still that this is almost at the acceptance threshold. The large focus on biology makes much of this paper harder to evaluate or appreciate.

---

> ### Author Response · Authors · 2018-11-26
> **We thank the reviewer for the emphasis on the multiple aspects of novelty present in neuron editing**
>
> We thank the reviewer for the emphasis on the multiple aspects of novelty present in neuron editing. We agree that both the method of neuron editing and the problem of out-of-sample extrapolation are both underrepresented in research currently, and hope our work can take a step towards addressing that.
>
> The reviewer seems mostly to have reservations with the first experiment, which was motivational in nature and not our main focus. We have significantly revised this experiment such that it addresses the reviewer’s critique. We hope that this revised version of the image experiment persuades the reviewer to consider a higher rating, matching that of the other two reviewers, as this appears to have been the (justifiably) main concern.
>
> We would like to address the additional feedback provided in-line:
>
> ---------------------------------
> “The fact that the editing function is fixed may greatly hinder its flexibility and applicability.
> In Section 3.1 and Figure 1, we are shown that NeuronEditting can turn images of horses with white backgrounds into images of horses with dark backgrounds (horses are an unseen class).
> NeuronEditting turns the horse darker as well. It seems that one could change the brightness and contrast of the image to obtain a similar effect, or one could or take the geometric mean of the image in [0,1] with the average target image and obtain a similar effect. Such traditional methods are also robust to unseen classes. Moreover, NeuronEditting's ability to change the brightness of the image is not that surprising given that brightness is some of the most basic image information. (In point of fact it is captured by the DC Coefficient, the very first coefficient from the discrete cosine transform which is used in JPEG.) What else can NeuronEditting do in the image domain? Can this be used to rotate or reflect MNIST digits? The biological experiments also appear to involve simple input transformations.“
> ---------------------------------
> We appreciate the feedback, but we emphasize that our goal is to learn a general, generic transformation on one dataset and apply it to others: not achieve some specific effect on images. Nonetheless, we have incorporated this helpful feedback and revised Experiment #1 in an updated version of the manuscript. We have now included a demonstration on CelebA, learning a transformation on males with black hair to males with blond hair, and then extrapolating to females with black hair. We show that neuron editing is able to isolate only the hair, and not change any other pixels in the image (like the horses in the previous version of the experiment).
>
> This obviously requires the network to have a disentangled internal representation, where a neuron edit can be done that selectively changes only hair pixels. We argue this cannot be done with a simple global transformation like subtracting the geometric mean, as the reviewer suggests.
>
> Moreover, as we point out in the manuscript, the fact that neuron editing is a piecewise linear function does not severely limit its expressivity, as powerful models like GANs can be implemented as piecewise linear functions with certain choices of activations, and the power of these models is obvious. We do not doubt there are other transformations that could achieve any specific desired goal, like using the geometric mean to change image brightness. However, these solutions are domain (image) and task (brightness) specific. We believe neuron editing is valuable as a general method for achieving any of these transformations with deep models, given their widespread use and applicability.
>
> We argue any reader would naturally assume that training a GAN for this task would succeed, which we demonstrate is not the case. Beyond providing this insight, we also introduce a way to achieve this goal within the framework of current deep architectures.
>
> In the end, our first experiment on images was a motivational introduction to the method. Images are visual and intuitive, thus making it easier to introduce the model with them. Our primary goals were with the biological applications and they are far more exciting use cases for neuron editing. The image example is the most tangible and thus introduces the ideas without entering into the details of the biology, so we think it still has value in the paper. But, the performance of other generative models on unseen cell types in the biological cases make them completely unusable, and as such neuron editing facilitates a learning task that is otherwise not possible.
>
> Concerning word2vec, we merely make the connection to help a reader who may be less familiar with the ideas around latent space manipulations, because word2vec is likely the most intuitive example. We also wanted to highlight why neuron editing is different, which may not be obvious to everyone. If the reviewer believes this detracts from the manuscript instead, we will happily remove it.

---

### Meta-Review · Area_Chair1 · 2018-12-16
**reasonable clarity and quality but unclear significance**

**Confidence:** 2
**Recommendation:** Reject

**Metareview:**

This was a borderline paper, as reviewers generally agreed that the method was a new method that was appropriately explained and motivated and had reasonable experimental results. The main drawbacks were that the significance of the method was unclear. In particular, the method might be too inflexible due to being based on a hard-coded rule, and it is not clear why this is the right approach relative to e.g. GANs with a modified training objective). Reviewers also had difficulty assessing the significance of the results on biological datasets. While such results certainly add to the paper, the paper would be stronger if the argument for significance could be assessed from more standard datasets.

A note on the review process: the reviewers initially scored the paper 6/6/6, but the review text for some of the reviews was more negative than a typical 6 score. To confirm this, I asked if any reviewers wanted to push for acceptance. None of the reviewers did (generally due to feeling the significance of the results was limited) and two of the reviewers decided to lower their scores to account for this.